# TapWeight: Reweighting Pretraining Objectives for Task-Adaptive Pretraining

**Ruiyi Zhang**  *ruz048@ucsd.edu*
*UC San Diego*

**Sai Ashish Somayajula**  *ssomayaj@ucsd.edu*
*UC San Diego*

**Pengtao Xie**  *p1xie@ucsd.edu*
*UC San Diego*

**Reviewed on OpenReview:** *https://openreview.net/forum?id=DCCw2CEVFS*

## Abstract

Large-scale general domain pretraining followed by downstream-specific finetuning has become a predominant paradigm in machine learning. However, discrepancies between the pretraining and target domains can still lead to performance degradation in certain cases, underscoring the need for task-adaptive continued pretraining (TAP). TAP methods typically involve continued pretraining on task-specific unlabeled datasets or introducing additional unsupervised learning objectives to enhance model capabilities. While many TAP methods perform continued pretraining with multiple pretraining objectives, they often determine the tradeoff parameters between objectives manually, resulting in suboptimal outcomes and higher computational costs. In this paper, we propose TapWeight, a task-adaptive pretraining framework which automatically determines the optimal importance of each pretraining objective based on downstream feedback. TapWeight reweights each pretraining objective by solving a multi-level optimization problem. We applied TapWeight to both molecular property prediction and natural language processing tasks, significantly surpassing baseline methods. Experimental results validate the effectiveness and generalizability of TapWeight. Our code is available at `https://github.com/ruz048/TapWeight`.

## 1 Introduction

Foundation models pretrained on large-scale general domain corpora have achieved state-of-the-art performance across a wide range of tasks (He et al., 2021a; Devlin et al., 2019; Brown et al., 2020). These models, which capture general knowledge for specific modalities such as text or images through unsupervised learning, are typically adapted to downstream tasks via finetuning. However, when there is a domain discrepancy between the pretraining corpus and the target task, direct finetuning of the pretrained model often fails to deliver optimal results (Lee et al., 2020; Chen et al., 2023; Xie et al., 2024). To address this challenge, downstream task-adaptive continued pretraining, or task-adaptive pretraining (TAP), has been introduced. TAP bridges this gap by introducing an additional continued pretraining stage between general domain pretraining and task specific finetuning. For example, Gururangan et al. (2020) conducts task-adaptive pretraining by performing unsupervised learning on the unlabeled data of the downstream task. Wu et al. (2021) introduces an additional perturbation masking objective during continued pretraining of a BERT model (Devlin et al., 2019), enhancing its performance on dialogue understanding tasks.

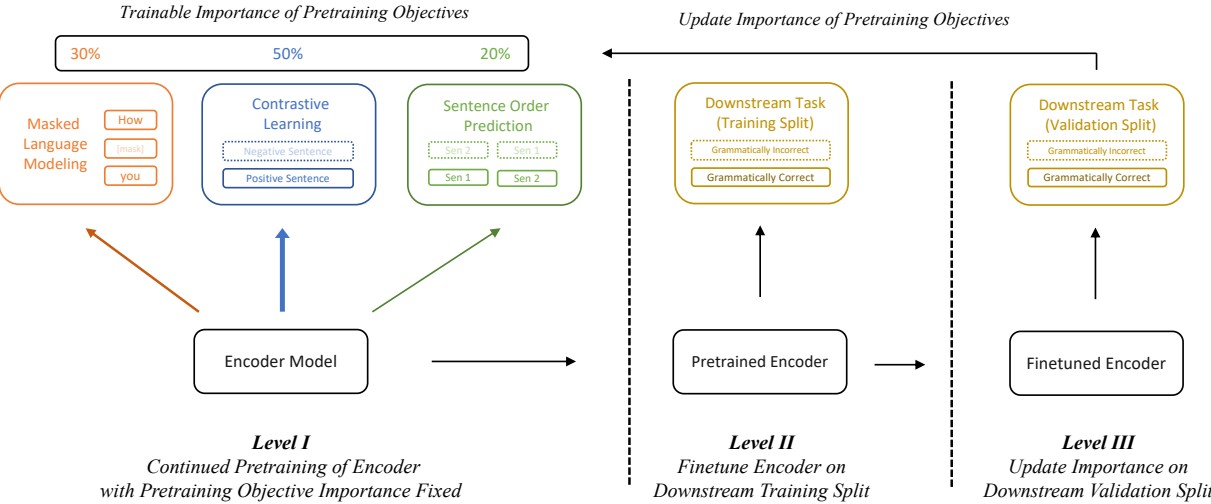

Figure 1: **An Overview of TapWeight.** In the first level, the model undergoes multi-objective pretraining with fixed tradeoff ratios between objectives. In the second level, the pretrained model is finetuned on the training split of the downstream dataset. In the third level, the finetuned model is evaluated on the validation split of the downstream dataset to compute a loss, and the trainable tradeoff parameters fixed in the first level are learned by minimizing this validation loss.

Among these, many existing task-adaptive pretraining methods consist of multiple pretraining objectives (Wu et al., 2021; Gao et al., 2021; Cui et al., 2023), making it challenging to determine the relative importance of each objective. Some TAP methods assign equal weight to each pretraining objective (Lee et al., 2020; Wu et al., 2021), disregarding their varying impact on downstream performance. For instance, Gao et al. (2021) shows that pretraining BERT with a contrastive learning (CL) objective results in better downstream performance on semantic textual similarity (STS) datasets than using masked language modeling (MLM) loss, indicating that the CL objective is more important than the MLM objective for these tasks. Other approaches attempt to manually tune the importance ratios through hyperparameter search (Gao et al., 2021), which often results in suboptimal performance and increased computational costs. This issue becomes particularly severe when the number of pretraining objectives is large, such as with the task-adaptive pretraining of a popular molecular model Imagemol, which involves 5 distinct pretraining objectives (Zeng et al., 2022). Raghu et al. (2021) propose a multi-level framework to learn the importance of pretraining objectives, with a focus on supervised pretraining. However, their approach faces challenges when scaling to large-scale unsupervised pretraining due to computational cost.

To address the aforementioned challenges, we propose a novel framework, TapWeight, designed to learn the optimal tradeoff parameters between various pretraining objectives during task-adaptive pretraining. The goal is to learn these optimal tradeoff parameters such that the pretrained model, after finetuning on a downstream task, achieves the best downstream task performance. Our approach involves a three-level optimization framework to learn these parameters. In the first level, we perform task-adaptive pretraining using initial tradeoff parameters, denoted as $\lambda$. These parameters are kept fixed in this level and will be updated in subsequent levels. The resulting pretrained model is thus a function of $\lambda$. In the second level, the pretrained model from the first level is finetuned on the training split of the downstream dataset. Consequently, the finetuned model becomes an implicit function of the tradeoff parameters. In the third level, the finetuned model is evaluated on the validation split of the downstream dataset, and the tradeoff parameters $\lambda$ are optimized by minimizing the validation loss. This end-to-end process allows the optimization problems in three levels to dynamically influence one another, forming an integrated framework that optimizes task-adaptive pretraining process and enhances downstream task performance. Moreover, TapWeight is broadly applicable to pretrained models with multiple pretraining objectives across various data modalities and downstream task types, demonstrating superior generalizability compared to existing

task-adaptive pretraining methods (Nishida et al., 2021; Cui et al., 2023). Figure 1 illustrates the complete framework of TapWeight.

We apply TapWeight for task-adaptive pretraining of a molecule representation model, Imagemol (Zeng et al., 2022), and language models including RoBERTa (Liu et al., 2019c) and DeBERTa (He et al., 2021b). Evaluating its performance across 13 molecular property prediction datasets and 11 natural language processing tasks, TapWeight significantly outperforms baseline methods across different tasks and model sizes. The superior performance of TapWeight highlights its effectiveness and generalizability. Our contribution can be summarized as follows:

- We propose TapWeight, an approach that automatically searches for the tradeoff parameters across multiple pretraining objectives and performs reweighted task-adaptive pretraining. TapWeight is formulated within a multi-level optimization (MLO) framework. We employ an efficient gradient descent algorithm to solve the MLO problem, obtaining the optimal tradeoff parameters for multiple pretraining objectives.

- We apply TapWeight for task-adaptive pretraining of a molecule representation model and a language model. Extensive experiments on 13 downstream datasets in molecular property prediction and 11 datasets in natural language processing underscore its effectiveness and generalizability.

## 2 Related Works

### 2.1 Domain / Task Adaptive Pretraining

To bridge the gap between general domain pretraining and downstream tasks in a specific domain, domain-adaptive pretraining (DAP) and task-adaptive pretraining (TAP) have been introduced (Gururangan et al., 2020). DAP performs continued pretraining on a large, unlabeled corpus from a similar domain as the downstream task. For example, BioBERT continues to pretrain a BERT model on a large-scale biomedical corpus, enhancing its performance on a variety of biomedical text mining tasks (Lee et al., 2020). Similarly, LegalBERT continues to pretrain a BERT model on legal documents to improve performance on legal NLP tasks (Chalkidis et al., 2020), while SciBERT leverages a large multi-domain corpus of scientific publications for further pretraining, enhancing its effectiveness on scientific NLP tasks (Beltagy et al., 2019). More recently, MEDITRON performs continued pretraining of a Llama-2 model with 80 billion parameters on text in medical domain, showing significant performance gains on major medical benchmarks (Chen et al., 2023). U-PaLM (Tay et al., 2023) performs continued pretraining on PaLM (Chowdhery et al., 2024) model with 540 billion parameters using UL2's mixture-of-denoiser pretraining objective (Tay et al., 2022), achieving performance improvement on many few-shot tasks, such as MMLU and GSM8K.

Although DAP significantly improves model performance on downstream tasks, it needs a large corpus of unlabeled data in a specific domain, which is not always available. To address this limitation, multiple task-adaptive pretraining (TAP) methods have emerged, which do not rely on additional domain-specific corpora beyond the downstream dataset itself. TAP methods can also be viewed as a novel finetuning process, where standard finetuning is preceded by low-cost continued pretraining. For instance, TAPT performs continued pretraining directly on the unlabeled training split of the downstream dataset (Gururangan et al., 2020). TAPTER first trains new word embeddings using the unlabeled training split of the downstream dataset, and then use these embeddings for continued pretraining of the model (Nishida et al., 2021). SimCSE introduces an additional constrastive learning loss in addition to the original masked language modelling loss to further pretrain a RoBERTa model, specifically enhancing its capability on standard semantic textual similarity tasks (Gao et al., 2021). PCP combines the idea of instruction tuning with conventional continued pre-training, consistently improving the performance of state-of-the-art prompt-based finetuning approaches on 21 benchmarks (Shi & Lipani, 2023). While existing TAP methods are effective, they are typically tailored to specific downstream tasks or data modalities (Wu et al., 2021; Cui et al., 2023). In contrast, TapWeight is applicable to pretrained models across diverse modalities and tasks with multiple pretraining objectives, underscoring its broad generalizability.

## 2.2 Multi-level Optimization

Many machine learning tasks can be formulated as multi-level optimization (MLO) problems, such as neural architecture search (Liu et al., 2019a; Chen et al., 2019; Xu et al., 2020), meta learning (Finn et al., 2017; Rajeswaran et al., 2019; Zhang et al., 2024), and hyperparameter optimization (Lorraine et al., 2020; Lorraine & Duvenaud, 2018; Mackay et al., 2019). MLO problems consist of multiple levels of optimization problems that are mutually dependent, making it challenging for common automatic differentiation algorithms to handle them. To tackle this challenge, multiple algorithms (Lorraine et al., 2020; Liu et al., 2019a; Rajeswaran et al., 2019) and libraries (Choe et al., 2023c;a) have been proposed to efficiently compute gradients in MLO problems.

Recently, MLO techniques have been widely adopted in data reweighting and task reweighting. In these methods, the weights of data or tasks are often treated as hyperparameters and optimized in the upper levels of MLO problems. For example, MetaWeightNet learns an explicit weighting function for each data point to maximize the performance on a small amount of unbiased meta-data (Shu et al., 2019). DoGE optimizes weights for each data domain using a small proxy model to guide the pretraining of larger models (Fan et al., 2024). MetaWeighting learns tradeoff parameters for each task in multi-task learning to minimize generalization loss (Mao et al., 2022). Raghu et al. (2021) proposes a multi-level optimization framework to learn the importance of supervised pretraining objectives based on feedback from downstream performance. Their method simulates the finetuning process by unrolling a few finetuning steps within one optimization level. Our method also falls within this category, with a specific focus on reweighting pretraining objectives for downstream task-adaptive continued pretraining. While our method shares a three-level optimization framework with Raghu et al. (2021) for identifying the importance of pretraining objectives, we adopt proximal regularization to simulate the finetuning process, reducing computational cost and enabling scalable unsupervised pretraining on large models.

## 2.3 Multi-task Learning

Multi-task learning (MTL) enables models to learn multiple tasks simultaneously, promoting knowledge sharing and transfer while mitigating task conflict. Existing MTL methods can be broadly categorized into two groups. The first group focuses primarily on architecture design for parameter sharing. For example, Ruder (2017) introduces two typical approaches for parameter sharing: hard sharing and soft sharing. Misra et al. (2016) proposes using trainable linear mappings to dynamically select different combinations of activation maps for different tasks. Rosenbaum et al. (2018) introduces a trainable router network to iteratively select functional blocks for different tasks. Liu et al. (2019b) employs soft attention modules to extract task-specific features from shared representations. Similarly, Yang et al. (2020) also utilizes a trainable router network, as in Rosenbaum et al. (2018), but with soft modularization that assigns probability weights to each connection between blocks.

The second group of MTL methods is based on optimization and gradient operations. For instance, Sener & Koltun (2018) formulates MTL as a multi-objective optimization problem, defining the overall objective as finding a Pareto-optimal solution. Chen et al. (2018) proposes a gradient normalization algorithm that dynamically adjusts the gradient magnitudes of each task. Yu et al. (2020) introduces gradient surgery, which projects a task's gradient onto the normal plane of another task's gradient if task conflicts exist. Mao et al. (2022) applies meta-learning to search for optimal task weights. Achituve et al. (2024) proposes the first Bayesian formulation for gradient aggregation in MTL and develops a new optimization algorithm based on posterior estimation.

Our method, TapWeight, shares more similarity with the second category as it also operates on gradients by reweighting each pretraining objective. However, TapWeight differs critically from all the MTL methods mentioned above, which generally assume that training and testing tasks are identical and do not consider a pretraining–finetuning scheme. In contrast, TapWeight specifically addresses the continued pretraining (CP) problem, where the MTL tasks during CP differ from those during finetuning. In this setting, optimizing solely for performance on pretraining tasks does not guarantee improved finetuning outcomes, making direct application of existing MTL methods inappropriate. To address this challenge, TapWeight introduces a

novel multi-level optimization framework that searches for pretraining task weights to maximize finetuning performance, effectively overcoming the limitations of previous MTL approaches in the CP context.

## 3 Method

### 3.1 Overview

Given $n$ continued pretraining objectives $\mathcal{T}_1, \mathcal{T}_2, ...\mathcal{T}_n$ and their corresponding training losses $\mathcal{L}_1, \mathcal{L}_2, ...\mathcal{L}_n$, we formulate the multi-objective continued pretraining loss $\mathcal{L}_{pt}$ as:

$$\mathcal{L}_{pt}(\theta, \lambda, \mathcal{D}_{pt}) = \sum_{i=1}^{n} \lambda_i \mathcal{L}_i(\theta, \mathcal{D}_{pt}) \tag{1}$$

where $\mathcal{D}_{pt}$ is the unsupervised pretraining dataset, $\theta$ denotes the pretraining model parameters, and $\lambda_i$ is the tradeoff parameter for each pretraining objective. We denote the target downstream task as $\mathcal{D}_{ft}$ and split it into $\mathcal{D}_{tr}, \mathcal{D}_{val}$ and $\mathcal{D}_{ts}$, which are training, validation and test splits respectively.

In our framework, TapWeight, we aim to automatically search for the optimal tradeoff weights $\lambda = \{\lambda_1, ..., \lambda_n\}$, so that the pretrained model achieves the highest performance on the test split $\mathcal{D}_{ts}$ after being fine-tuned on the downstream dataset $\mathcal{D}_{tr}$. To achieve this, our method consists of three levels of optimization problems. In the first level, we perform continued pretraining of the model, with tradeoff weights tentatively fixed. In the second level, we conduct finetuning of the pretrained model on the training split of the downstream dataset. In the third level, we compute a loss by applying the finetuned model on the validation split of the downstream dataset, and optimize the tradeoff parameters by minimizing this loss. We next formally define these three levels under a multi-level optimization framework.

### 3.2 TapWeight Framework

**Level I** In the first level, we aim to perform continued pretraining for the model. Formally, the optimization problem (OP) is to optimize the model weights $\theta$ to minimize the multi-objective pretraining loss $\mathcal{L}_{pt}$ on a unlabeled dataset $\mathcal{D}_{pt}$:

$$\theta^*(\lambda) = \arg\min_{\theta} \mathcal{L}_{pt}(\theta, \lambda, \mathcal{D}_{pt}) \tag{2}$$

Since the optimal solution $\theta^*$ to this problem depends on the value of the tradeoff parameter, it is an implicit function of $\lambda$, denoted as $\theta^*(\lambda)$.

**Level II** In the second level, we aim to finetune the pretrained model with optimal parameters $\theta^*$ obtained from previous level on the downstream dataset. Raghu et al. (2021) formulates the optimization problem using the same set of parameters $\theta$ at the lower level and differentiates through the entire gradient update trajectory, which imposes relatively high computational and memory burdens. In contrast, optimizing distinct sets of parameters at different levels enables the use of implicit differentiation methods, which significantly reduces computational costs, as detailed in Section 3.3. Therefore, we create a model with new parameters $\omega$ that are different from those in the pretrained model, but with a regularization loss $\mathcal{R}$ between $\omega$ and $\theta$ to encourage them to be close, inspired by Rajeswaran et al. (2019). This proximal constraint casts strong dependence between $\omega^*$ and $\theta^*$, closely resembling the real finetuning process. Formally, the OP in this level is to optimize $\omega$ by minimizing the weighted summation of finetuning loss $\mathcal{L}_{tr}$ and the proximal regularization loss $\mathcal{R}$:

$$\omega^*(\theta^*(\lambda)) = \arg\min_{\omega} \mathcal{L}_{tr}(\omega, \mathcal{D}_{tr}) + \gamma \mathcal{R}(\omega, \theta^*(\lambda)) \tag{3}$$

where $\mathcal{D}_{tr}$ is the training split of the downstream dataset, and $\gamma$ is a tradeoff hyperparameter to balance the finetuning loss and regularization loss. In practice, we select the mean squared error (MSE) loss as the regularization loss $\mathcal{R}$, and we further elaborate the reason for this choice in Appendix E. The optimal solution of $\omega$ in this level is a function of $\theta^*$ due to the loss term $\mathcal{R}$, which is in turn a function of $\lambda$, denoted as $\omega^*(\theta^*(\lambda))$.

**Level III**   In the third level, we aim to search for the optimal tradeoff parameters $\lambda^*$ between pretraining objectives. Formally, the OP in this level is to optimize $\lambda$ to minimize the validation loss $\mathcal{L}_{val}$:

$$\min_\lambda \mathcal{L}_{val}(\omega^*(\lambda), \mathcal{D}_{val}) \tag{4}$$

where $\mathcal{D}_{val}$ is the validation split of the downstream dataset. In practice, we reparameterize each tradeoff weight $\lambda_i$ via a softmax over unconstrained variables $\beta$:

$$\lambda_i = \frac{\exp(\beta_i)}{\sum_{j=1}^n \exp(\beta_j)}.$$

We optimize $\beta$ rather than $\lambda$ directly. This both guarantees tradeoff weights to be non-negative ($\lambda_i \geq 0$) and imposes a constraint on the sum ($\sum_{i=1}^n \lambda_i = 1$) automatically.

**Multi-level Optimization Framework**   In this way, we formulate a three-level optimization problem with OPs in different levels mutually dependent on each other:

$$\min_\lambda \mathcal{L}_{val}(\omega^*(\lambda), \mathcal{D}_{val}) \tag{5}$$
$$s.t. \quad \omega^*(\theta^*(\lambda)) = \arg\min_\omega \mathcal{L}_{tr}(\omega, \mathcal{D}_{tr}) + \gamma\mathcal{R}(\omega, \theta^*(\lambda))$$
$$\theta^*(\lambda) = \arg\min_\theta \mathcal{L}_{pt}(\theta, \lambda, \mathcal{D}_{pt})$$

By solving this multi-level optimization problem, we are able to reweight each continued pretraining objective based on feedback from validation performance on downstream tasks. In practice, both $\theta$ and $\omega$ in Equation 5 are initialized with model weights from general-domain pretraining.

### 3.3 Optimization Algorithm

In this section, we illustrate the algorithm we use to efficiently approximate the gradient of loss $\mathcal{L}_{val}$ in the third level with respect to the tradeoff parameter $\lambda$. This full derivative $\frac{d\mathcal{L}_{val}}{d\lambda}$ can be computed with the following equation using chain rule:

$$\frac{d\mathcal{L}_{val}}{d\lambda} = \frac{\partial\mathcal{L}_{val}}{\partial\omega^*} \times \frac{\partial\omega^*}{\partial\theta^*} \times \frac{\partial\theta^*}{\partial\lambda} \tag{6}$$

In the right hand side of Equation 6, the green term, a partial derivative vector, can be directly computed with popular automatic differentiation libraries, such as Pytorch (Paszke et al., 2019). However, directly computing the two red terms, which are best-response Jacobian matrices, can be computationally prohibitive due to the lack of analytical solutions to these optimization problems. Inspired by previous works (Lorraine et al., 2020; Zhang et al., 2021), we use Implicit Function Theorem (IFT) based methods to approximate the best-response Jacobian matrices. We include more details of IFT based gradient computation method in Appendix A. In this way, we are able to compute both red terms in Equation 6 efficiently, thereby obtaining the gradient of $\mathcal{L}_{val}$ with respect to $\lambda$. We then optimize the tradeoff parameter $\lambda$ with gradient descent. The complete algorithm is implemented using the Betty library (Choe et al., 2023c;b). We present the complete optimization algorithm of TapWeight in Algorithm 1.

---

**Algorithm 1** TapWeight-optimization

---

**Input:** Unsupervised pretraining dataset $\mathcal{D}_{pt}$, training dataset $\mathcal{D}_{tr}$, validation dataset $\mathcal{D}_{val}$, total global optimization steps $M$

1: **for** $i = 1, \ldots, M$ **do**
2:      Sample mini-batches: $X_{pt} \sim \mathcal{D}_{pt}$, $X_{tr} \sim \mathcal{D}_{tr}$, $X_{val} \sim \mathcal{D}_{val}$
3:      Compute gradient $\nabla_\theta \mathcal{L}_1(\theta, \lambda; X_{pt})$, where $\mathcal{L}_1 = \mathcal{L}_{pt}(\theta, \lambda, X_{pt})$ (Equation 2)
4:      Update: $\theta \leftarrow \theta - \alpha_1 \nabla_\theta \mathcal{L}_1$
5:      Compute hyper-gradient $\dfrac{d\mathcal{L}_2}{d\omega}$, where $\mathcal{L}_2 = \mathcal{L}_{tr}(\omega, X_{tr}) + \gamma \, \mathcal{R}(\omega, \theta^*(\lambda))$ (Equation 3)
6:      Update: $\omega \leftarrow \omega - \alpha_2 \dfrac{d\mathcal{L}_2}{d\omega}$
7:      Compute hyper-gradient $\dfrac{d\mathcal{L}_3}{d\lambda}$, where $\mathcal{L}_3 = \mathcal{L}_{val}(\omega^*(\lambda), X_{val})$ (Equation 4)
8:      Update: $\lambda \leftarrow \lambda - \alpha_3 \dfrac{d\mathcal{L}_3}{d\lambda}$
9: **end for**

**Output:** Optimal weights $\theta^*$, $\omega^*$, $\lambda^*$

---

## 4 Experiments

### 4.1 Molecular Property Prediction

In this section, we use TapWeight for task-adaptive pretraining of molecular image models and validate the effectiveness of our framework on the downstream task of molecular property prediction.

#### 4.1.1 Preliminary

Given a large unlabeled molecular dataset $\mathcal{D} = \{x_i\}_{1 \leq i \leq n}$ containing millions of molecules, we define a multi-objective continued pretraining loss inspired by Imagemol (Zeng et al., 2022):

$$
\begin{aligned}
\mathcal{L}(x) = &\lambda_1 \mathcal{L}_{mg1}(x) + \lambda_2 \mathcal{L}_{mg2}(x) + \lambda_3 \mathcal{L}_{mg3}(x) \\
&+ \lambda_4 \mathcal{L}_{jpp}(x) + \lambda_5 \mathcal{L}_{mcl}(x)
\end{aligned} \tag{7}
$$

where $x$ represents a molecular image, and $\lambda = \{\lambda_i\}_{1 \leq i \leq 5}$ are tradeoff parameters. $\mathcal{L}_{mg1}$, $\mathcal{L}_{mg2}$, and $\mathcal{L}_{mg3}$ are MACCS key (Durant et al., 2002) clustering-based classification losses with different number of clusters. $\mathcal{L}_{jpp}$ is a jigsaw puzzle prediction loss, where the model solves a jigsaw puzzle on the same molecular image. $\mathcal{L}_{mcl}$ is a mask-based contrastive learning loss, which generates constrastive pairs by masking molecular images. Details of these pretraining objectives can be found in Appendix C.1.

The multi-objective loss $\mathcal{L}$ is optimized on the complete unlabeled dataset $\mathcal{D}$ to train a molecular image encoder. The learnt encoder can be further finetuned on downstream datasets for various molecular tasks. Existing approaches typically set the tradeoff parameters $\lambda$ equally across different pretraining objectives, overlooking the varying contributions of each objective to specific downstream tasks (Zeng et al., 2022). We address this challenge by applying TapWeight framework for continued pretraining of the molecular image encoder.

#### 4.1.2 Experimental Settings

We perform continued pretraining of a pretrained Imagemol model on a dataset $\mathcal{D}$, consisting of 1 million molecules from PubChem (Kim et al., 2023). For downstream tasks, we employ the MoleculeNet benchmark, which includes 8 classification datasets focused on predicting biophysical and physiological properties essential for drug discovery (Wu et al., 2017). We generate the training, validation and test split of these

| Method | BACE | BBBP | ClinTox | Sider | Tox21 | ToxCast | HIV | MUV | Avg. |
|---|---|---|---|---|---|---|---|---|---|
| Dataset Size | 1,513 | 2,039 | 1,478 | 1,427 | 7,831 | 8,575 | 41,127 | 93,087 | |
| AttrMask | 77.2 | 70.2 | 68.6 | 60.4 | 74.2 | 62.5 | 74.3 | 73.9 | 70.2 |
| ContextPred | 78.6 | **71.2** | 73.7 | 59.3 | 73.3 | 62.8 | 75.8 | 72.5 | 70.9 |
| GraphMVP | 76.8 | 68.5 | 79.0 | 62.3 | 74.5 | 62.7 | 74.8 | 75.0 | 71.7 |
| Finetuning (Imagemol) | 80.1 | 67.3 | 78.5 | 63.6 | 76.5 | 65.4 | 75.6 | 78.4 | 73.2 |
| **TapWeight (ours)** | **83.1** | **71.2** | **81.3** | **64.5** | **77.0** | **66.1** | **78.4** | **80.5** | **75.3** |

Table 1: Results of molecular property prediction on 8 classification tasks in MoleculeNet benchmark, in terms of AUROC. Higher values are better for all results, and the best results are shown in **bold**.

downstream datasets by applying scaffold splitting [1] with an 8:1:1 ratio. We use AUROC as the evaluation metric for all classification datasets, MAE for Qm7 and Qm9 datasets, and RMSE for all other regression datasets. In addition to Imagemol, we benchmark against Graph Neural Network (GNN)-based molecular property prediction methods, including pretraining approaches such as attribute masking, context prediction Hu et al. (2020), and GraphMVP (Liu et al., 2022). The pretrained molecular image encoder is based on a ResNet18 model, with the final classification layer removed (He et al., 2015). More detailed descriptions are provided in the Appendix for the datasets (C.2), baselines (C.3), and hyperparameter settings (C.4).

### 4.1.3   Results

Table 1 show the results of various methods across 8 molecular property classification tasks from MoleculeNet benchmark. Our method outperforms all baseline methods on all 8 datasets, showcasing the effectiveness of our method. On average, our method achieves an AUROC of 75.3, compared to 73.2 for the Imagemol model without continued pretraining. Similarly, Table 2 displays the results for 5 regression tasks in the MoleculeNet benchmark, where our method once again surpasses all baselines on each task. Experimental results validate the effectiveness of our method on both classification and regression tasks. Specifically, the superior performance of our method over Imagemol validates the necessity of downstream-guided continued pretraining following general pretraining. Notably, our method consistently outperforms baseline approaches regardless of the size of the finetuning dataset, demonstrating the robustness of our approach. It is worth mentioning that TAPT (Gururangan et al., 2020) is not applicable to this task, as clustering-based losses, such as $\mathcal{L}_{mg3}$, are not well-suited for direct application on small unlabeled datasets where the number of data points is smaller than the predefined number of clusters. In contrast, TapWeight does not face such limitations, demonstrating its generalizability.

## 4.2   Natural Language Processing

In this section, we validate the effectiveness of TapWeight for continued pretraining of a masked language model (MLM) with its application to natural language processing tasks.

### 4.2.1   Prelimimary

Given a large-scale raw-text dataset $\mathcal{D} = \{x_i\}_{1 \le i \le n}$ consisting of millions of sentences, we define the following continued pretraining loss:

| Method | Freesolv | Esol | Lipo | Qm7 | Qm9 |
|---|---|---|---|---|---|
| Dataset Size | 642 | 1,128 | 4,200 | 6,830 | 133,885 |
| AttrMask | 2.95 | 1.37 | 0.81 | 161.7 | 5.03 |
| ContextPred | 3.01 | 1.35 | 0.83 | 153.2 | 4.95 |
| GraphMVP | 2.21 | 1.13 | 0.79 | 134.5 | 4.76 |
| Finetuning (Imagemol) | 3.04 | 1.11 | **0.76** | 141.0 | 4.52 |
| **TapWeight (ours)** | **1.91** | **1.06** | **0.76** | **126.0** | **4.28** |

Table 2: Results of molecular property prediction on 5 regression tasks in MoleculeNet benchmark. Lower values are better for all results, and the best results are shown in **bold**.

---

[1]Scaffold splitting partitions molecules based on their core structural frameworks to ensure that structurally similar compounds do not appear across training and test sets, thereby providing a more rigorous evaluation of model generalization.

$$\mathcal{L}(x) = \lambda_1 \mathcal{L}_{mlm}(x) + \lambda_2 \mathcal{L}_{cl}(x) + \lambda_3 \mathcal{L}_{sop}(x) \tag{8}$$

where $x$ is a sentence, and $\lambda = \{\lambda_i\}_{1 \leq i \leq 3}$ are tradeoff parameters. $\mathcal{L}_{mlm}$ represents the masked language model loss, which involves randomly masking tokens in the input sentences and predicting these masked tokens (Devlin et al., 2019). $\mathcal{L}_{cl}$ denotes the contrastive learning loss, where an input sentence is used to predict itself with standard dropout applied as noise (Gao et al., 2021). $\mathcal{L}_{sop}$ is the sequence ordering prediction loss, which emphasizes inter-sentence conherence (Lan et al., 2020). We include details of these losses in Appendix D.1.

The multi-objective loss $\mathcal{L}$ is optimized on the raw text dataset $\mathcal{D}$ for continued pretraining of a Transformer encoder. The learnt encoder can then be finetuned on downstream NLP datasets. In existing works (Gao et al., 2021), the tradeoff parameters $\lambda$ for different pretraining objectives require manual hyperparameter tuning, which is time-consuming and often leads to suboptimal results. We address this challenge by applying TapWeight for the continued pretraining of a Transformer encoder, enabling the automatic determination of the importance for each objective.

### 4.2.2 Experimental Settings

We perform continued pretraining of masked language models on a raw-text dataset $\mathcal{D}$ consisting of 1 million sentences from Wikipedia (Gao et al., 2021). For downstream evaluation, we use RCT (Dernoncourt & Lee, 2017), AGNews (Zhang et al., 2015) and IMDB (Maas et al., 2011) datasets, which are widely used for evaluation of TAP methods (Gururangan et al., 2020; Shi & Lipani, 2023). We also use the GLUE benchmark for evaluation, which comprises 8 natural language understanding tasks, including sentiment analysis, semantic similarity prediction, and grammaticality classification (Wang et al., 2019). We use Matthew's Correlation for the CoLA dataset, Pearson/Spearman Correlation for the STS-B dataset, and accuracy for all other datasets. Our baseline methods are based on RoBERTa (Liu et al., 2019c) and DeBERTa (He et al., 2021b) models, including direct finetuning, TAPT based continued pretraining, PCP based continued pretraining (Shi & Lipani, 2023), and SimCSE based continued pretraining. More detailed descriptions are provided in the Appendix for the datasets (D.2), baselines (D.3), and hyperparameter settings (D.4).

### 4.2.3 Results

Table 3 reports the performance of various methods on three datasets: RCT, AGNews, and IMDB, evaluated using, RoBERTa-base (125M parameters), RoBERTa-large (355M parameters) and DeBERTa-xlarge (750M parameters). The RCT dataset involves classifying sentences in biomedical texts based on their functional roles, AGNews focuses on topic classification of news articles, and IMDB is a dataset for sentiment analysis of movie reviews. The results show that TapWeight consistently outperforms baseline methods across all 3 tasks and all 3 model sizes. These findings validate the robustness of TapWeight across diverse domains (biomedical, news, and reviews), while also highlighting its scalability across different model sizes. Moreover, our method surpasses the SimCSE method on all 3 tasks, showcasing the effectiveness of reweighting pretraining objectives, as SimCSE uses a fixed ratio between

| Method | RCT | AGNews | IMDB |
|---|---|---|---|
| Dataset Size | 78,387 | 127,600 | 50,000 |
| Finetuning ($R_b$) | 86.3 | 93.2 | 94.5 |
| SimCSE ($R_b$) | 85.9 | 93.0 | 94.1 |
| TAPT ($R_b$) | 86.4 | 93.5 | 94.7 |
| **TapWeight ($R_b$)** | **86.7** | **93.8** | **95.1** |
| Finetuning ($R_l$) | 86.9 | 94.0 | 95.2 |
| SimCSE ($R_l$) | 86.5 | 93.8 | 95.0 |
| TAPT ($R_l$) | 86.9 | 94.2 | 95.1 |
| **TapWeight ($R_l$)** | **87.4** | **94.8** | **95.5** |
| Finetuning ($D_{xl}$) | 87.5 | 94.7 | 95.4 |
| SimCSE ($D_{xl}$) | 87.0 | 93.9 | 94.7 |
| TAPT ($D_{xl}$) | 87.7 | 94.1 | 95.4 |
| **TapWeight ($D_{xl}$)** | **87.9** | **95.0** | **96.1** |

Table 3: Results of RoBERTa-base ($R_b$), RoBERTa-large ($R_l$) and DeBERTa-xlarge ($D_{xl}$) on RCT, AGNews and IMDB datasets in terms of accuracy. Higher values are better for all results, and the best results are shown in **bold**.

| Method | MNLI | QNLI | QQP | RTE | SST | MRPC | CoLA | STSB | Avg. |
|---|---|---|---|---|---|---|---|---|---|
| Dataset Size | 392,702 | 104,743 | 363,871 | 2,490 | 67,349 | 3,668 | 8,551 | 5,749 | |
| Finetuning | 86.7 | **92.8** | 90.3 | 77.8 | 94.8 | 89.3 | 61.6 | **91.2** | 85.6 |
| SimCSE | 85.6 | 90.1 | 90.7 | 74.6 | 91.1 | 89.2 | 59.7 | 91.0 | 83.6 |
| TAPT | 85.2 | 91.3 | 90.2 | 78.2 | 93.7 | 90.1 | 61.5 | 90.9 | 85.1 |
| PCP | 86.5 | 91.5 | 90.6 | 80.1 | 93.9 | 89.8 | 61.2 | **91.2** | 85.6 |
| **TapWeight (ours)** | **86.8** | 92.5 | **91.1** | **80.7** | **94.9** | **90.2** | **62.3** | **91.2** | **86.2** |

Table 4: Results of different methods in GLUE benchmark. All methods are applied to a RoBERTa-base model. Higher values are better for all results, and the best results are shown in **bold**.

MLM and CL losses during continued pretrain-
ing. Additionally, TapWeight outperforms the RoBERTa+TAPT approach, demonstrating that our strategy of leveraging downstream datasets by reweighting pretraining objectives is more effective than simply pretraining the model with unlabeled downstream data, as TAPT does.

Table 4 presents the results of various methods on 8 natural language understanding tasks from the GLUE benchmark. TapWeight consistently outperforms all baseline methods across all 8 datasets, showcasing the effectiveness of our method on tasks other than the 3 datasets mentioned above. On average, our method achieved a score of 86.2, while finetuning a RoBERTa model without continued pretraining only got 85.6. Furthermore, TapWeight outperforms the RoBERTa+PCP approach, further underscoring its effectiveness and advantage compared to popular TAP methods. To sum up, the superior performance on both molecule property prediction and natural language understanding highlights the generalizability of our method across multiple data modalities and downstream tasks.

We further applied TapWeight to a more recent pretrained LM, DeBERTa-v3-large (He et al., 2023), to validate the generalizability of TapWeight. Specifically, we compared TapWeight to direct finetuning of DeBERTa-v3-large on both the CoLA dataset and the MRPC dataset from the GLUE benchmark, with the results shown in Table 5. We observe that TapWeight still clearly outperforms the baseline method on both datasets, highlighting its generalizability to more recent pretrained LMs.

| Dataset | CoLA | MRPC |
|---|---|---|
| Finetuning | 73.0 | 91.5 |
| **TapWeight (ours)** | **74.1** | **92.1** |

Table 5: Results of TapWeight and Finetuning using DeBERTa-v3-large on CoLA and MRPC datasets.

## 4.3 Ablation Studies

In this section, we perform ablation studies to evaluate the effectiveness of individual components within our framework. All experiments are conducted on the classification tasks in the molecular property prediction benchmark.

**Pretraining Objective Reweighting**  We validate the effectiveness of our pretraining objective reweighting strategy by comparing our method to continued pretraining with a fixed importance for each objective. As shown in Table 6, our method outperforms this baseline (CP w/o Reweighting) across all datasets, demonstrating the advantage of dynamically reweighting pretraining objectives in the continued pretraining process.

**Multi-level Optimization**  We validate the effectiveness of the multi-level (tri-level) optimization (MLO) framework by reducing our method to a bi-level optimization (BLO) (Xie, 2023) based method. Specifically, we merge the first and second level of problems from the TapWeight framework to form the lower-level problem in the new BLO baseline, where the model is optimized jointly using both the unsupervised pretraining loss on the unlabeled continued pretraining dataset $\mathcal{D}_{pt}$ and the finetuning loss on the training split of the downstream dataset $\mathcal{D}_{tr}$. In the upper-level problem, the importance for each pretraining objective is learned using the validation split of the downstream dataset. Formally, we define the following BLO problem:

| Method | BACE | BBBP | ClinTox | Sider | Tox21 | ToxCast | HIV | MUV | Avg. |
|---|---|---|---|---|---|---|---|---|---|
| CP w/o Reweighting | 78.8 | 66.1 | 77.4 | 60.3 | 74.6 | 62.7 | 76.9 | 71.6 | 71.1 |
| TapWeight w/o MLO | 83.0 | 68.5 | 79.5 | 63.5 | 76.3 | 65.9 | 77.2 | 77.3 | 73.9 |
| **TapWeight** | **83.1** | **71.2** | **81.3** | **64.5** | **77.0** | **66.1** | **78.4** | **80.5** | **75.3** |

Table 6: **Ablation Studies.** Results of molecular property classification using our method and baseline methods, in terms of AUROC. Higher values are better for all results, and the best results are shown in **bold**.

$$\min_{\lambda} \mathcal{L}_{val}(\theta^*(\lambda), \mathcal{D}_{val}) \tag{9}$$
$$s.t. \quad \theta^*(\lambda) = \arg\min_{\theta} \mathcal{L}_{pt}(\theta, \lambda, \mathcal{D}_{pt}) + \gamma \mathcal{L}_{tr}(\theta, \mathcal{D}_{tr})$$

However, optimizing these two types of losses in the lower level requires extensive tuning of the tradeoff parameters $\gamma$, and often leads to competition between losses which results in performance decrease. As shown in Table 6, our MLO based reweighting method outperforms the BLO based approach across all datasets, highlighting the advantage of formulating multiple optimization problems. Nevertheless, BLO method still outperforms the baseline continued pretraining methods with fixed tradeoff parameters, indicating the necessity of using reweighting strategies.

**CP with Learned Final Weights** We conducted additional experiments on three MoleculeNet datasets (Bace, Clintox, and HIV) to evaluate the performance of continued pretraining (CP) using the tradeoff weights identified by TapWeight (CP-final-weights), with results shown in Table 7. On average, this baseline achieved a score of 80.6, which is comparable to TapWeight's score of 80.9, demonstrating the effectiveness of the tradeoff weights found by TapWeight. However, this approach incurs additional computational cost due to an extra round of CP, without yield-

| Dataset | BACE | Clintox | HIV |
|---|---|---|---|
| CP-final-weights | 82.3 | 81.1 | **78.5** |
| **TapWeight (ours)** | **83.1** | **81.3** | 78.4 |

Table 7: Performance of continued pretraining using the tradeoff weights previously identified by TapWeight (CP-final-weights) on three MoleculeNet datasets.

ing significant performance gains. Therefore, it is more practical to directly use the model weights obtained from TapWeight without conducting an additional CP stage.

### 4.4 Qualititive Analysis

In this section, we present the evolution trend of the pretraining objective weights along the training trajectory using our method. As shown in Figure 2, we plot the value of $\lambda$ for 3 regression tasks (Esol, Lipo, Freesolv) and 3 classification tasks (Tox21, Toxcast, Clintox) with respect to the global training step. Our observations reveal that different downstream datasets require varying importance for each pretraining objective. For example, the JPP pretraining objective, $\mathcal{L}_{jpp}$, plays an key role in Lipo and Toxcast datasets, whereas the MG3 pretraining objective, $\mathcal{L}_{mg3}$, is more critical for Esol, Freesolv and Tox21 datasets. The diverse requirements of pretraining objectives across downstream datasets emphasize the need for a reweighting method like TapWeight, providing a clear explanation for why our method outperforms baseline approaches. Furthermore, similar downstream tasks exhibit some degree of similarity in the weights assigned to pretraining tasks. For instance, the Esol and Freesolv datasets, both focused on predicting physical chemistry properties of molecules, assign large weights to the MG3 pretraining objective. In contrast, the ToxCast and ClinTox datasets, which involve predicting molecular toxicity, assign smaller weights to the MG3 objective.

### 4.5 Computation Cost

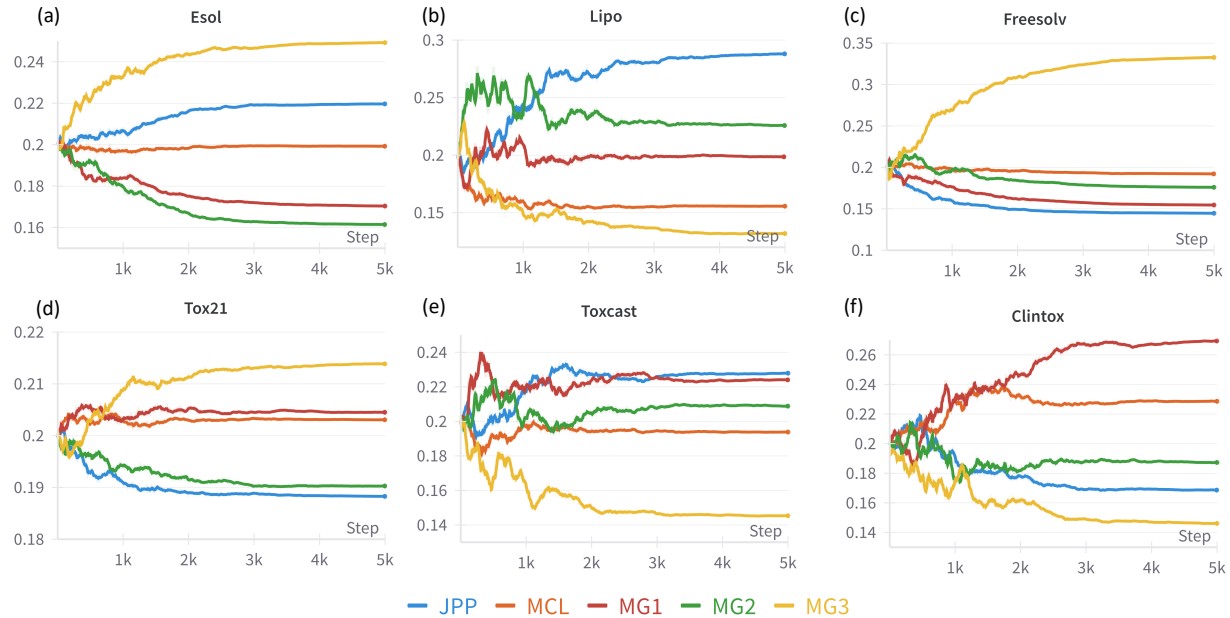

Figure 2: Evolution of the tradeoff parameter $\lambda$ over the training steps of TapWeight on the following downstream datasets: (a) Esol, (b) Lipo, (c) Freesolv, (d) Tox21, (e) Toxcast, and (f) Clintox.

| Dataset | FT | CP+FT | TAPT | PCP | TapWeight |
|---------|-----|-------|------|------|-----------|
| QQP | ×1 | ×2.54 | ×1.57 | ×2.26 | ×3.76 |

Table 8: Training cost of baseline methods and our method TapWeight on QQP dataset.

In this section, we compare the training time (wall time) of our method with baseline methods on the QQP, MUV and Qm9 datasets, as shown in Table 8 and Table 9. We use finetuning (FT), continued pretraining with a fixed tradeoff ratio (CP+FT), TAPT and PCP as baselines, normalizing the time cost of FT as 1. While TapWeight results in an increase in training time compared to baseline methods, its substantial improvement across multiple downstream tasks generally justifies the additional cost. However, in real-world

| Dataset | FT | CP+FT | TapWeight |
|---------|-----|-------|-----------|
| MUV | ×1 | ×2.18 | ×3.29 |
| Qm9 | ×1 | ×2.76 | ×3.93 |

Table 9: Training cost of baseline methods and our method TapWeight on MUV and Qm9 datasets.

applications where training time is a critical factor, TapWeight may not be the ideal choice, representing a limitation of our approach.

## 5 Conclusion and Future Work

In this paper, we propose a task-adaptive continued pretraining method that dynamically reweights each pretraining objective within a multi-level optimization framework. Experiments in both molecule property prediction and natural language processing validate the effectiveness and generalizability of our method. Given the success of TapWeight, several promising future research directions emerge. For instance, large multimodal pretrained models have recently gained popularity (Liu et al., 2023; Zhu et al., 2024). The combination of multiple modalities introduces a greater number of potential continued pretraining objectives, presenting necessities of applying TapWeight in this context.

## 6 Acknowledgement

This research was supported by NSF IIS2405974, NSF IIS2339216, NIH R35GM157217, and NIH R21GM154171.

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

## A  Optimization Algorithm

In this section, we give an example to briefly illustrate how to use Implicit Function Theorem (IFT) to compute best-response Jacobian matrices. Take $\frac{\partial \theta^*}{\partial \lambda}$ term in Equation 6 as an example: although $\theta^*$ is an implicit function of $\lambda$, the exact value of $\theta^*(\lambda)$ given a value of $\lambda$ is usually approximated with gradient descent algorithms. As there is no analytical solution of $\theta^*(\lambda)$, it is difficult to directly compute the gradient $\frac{\partial \theta^*}{\partial \lambda}$. To tackle this challenge, we compute this gradient using IFT following previous literature (Lorraine et al., 2020):

$$\frac{\partial \theta^*}{\partial \lambda} = -[\nabla^2 \mathcal{L}_{pt}(\theta)]^{-1} \times \frac{\partial^2 \mathcal{L}_{pt}}{\partial \theta \, \partial \lambda^T} \tag{10}$$

The green term, a second-order mixed partial derivative matrix, can be directly computed using automatic differentiation. Nevertheless, directly computing the red term, which is the invert of a Hessian matrix $\nabla^2 \mathcal{L}_{pt}(\theta)$, is computational expensive due to its $O(n^3)$ complexity. Various methods have been proposed to approximate the inverted Hessian matrix, including Neumann series (Lorraine et al., 2020), conjugate gradients (Rajeswaran et al., 2019) and finite difference (Zhang et al., 2021). In TapWeight, we select finite difference as the approximation method, thus enabling efficient computation of best-response Jacobian matrices.

## B  Complexity Analysis of TapWeight

In this section, we discuss the time complexity of TapWeight. In summary, our optimization algorithm scales linearly with the number of model parameters, owing to the approximation of the best-response Jacobian matrix-vector multiplication, inspired by Lorraine et al. (2020) and Zhang et al. (2021). We outline our analysis below:

Assume the number of optimizable parameters in the model is $n$. Directly computing the full derivative in Equation 6 would incur a complexity of $O(n^3)$, because the best-response Jacobian matrix requires computing an inverse Hessian-vector product (iHVP), and directly inverting a Hessian itself is $O(n^3)$. To address this, we leverage the Neumann series approximation to reduce the complexity to $O(n^2)$, using the following formula inspired by paper Lorraine et al. (2020):

$$v \cdot \left( \nabla_w^2 \mathcal{L}(w) \right)^{-1} \approx v \cdot \sum_{j=0}^{M} \left( I - \nabla_w^2 \mathcal{L}(w) \right)^j$$

where $M$ is a hyperparameter in practice. However, if we directly compute the Hessian above, the resulting quadratic complexity is still computationally intensive for models with billions of parameters. Inspired by paper Zhang et al. (2021), we further reduce the complexity by leveraging the Hessian-vector product (HVP) and finite differences. HVP has a complexity of $O(n)$ as implemented in modern automatic differentiation (AD) libraries (e.g., `torch.autograd.functional.hvp` in Pytorch (Paszke et al., 2019)), which is more efficient than directly computing the Hessian. A detailed algorithm for computing the best-response Jacobian-vector product can be found in Appendix B ("Practical Implementation of Hypergradient") of Zhang et al. (2021), where only a limited number of first-order gradient computations are required. Since gradient computation is $O(n)$ in modern AD libraries, the total complexity of multiplying the first two terms in Equation 6 is also $O(n)$. The resulting vector is then multiplied by a subsequent term, which can also be resolved similarly in $O(n)$ time. Overall, computing the right-hand side (RHS) of Equation 6 has a complexity of $O(n)$.

Other components of our optimization algorithm—including loss computation, gradient backpropagation, and parameter updates—also operate in $O(n)$ time. Therefore, the total complexity of the TapWeight optimization algorithm grows linearly with the model size $n$.

# C  Molecule Property prediction

## C.1  Pretraining Objectives

We use 3 types of pretraining objectives for continued pretraining of an Imagemol model (Zeng et al., 2022) to enhance its performance on molecule property prediction tasks.

**Multi-Granularity Clustering**   In this pretraining objective, we first perform K-means clustering to the unlabeled training dataset of molecules using their chemical structural fingerprint. After clustering, each molecule is assigned with a pseudo-label, and the molecular encoder model is pretrained by predicting this label. Formally,

$$\mathcal{L}_{mg1} = \sum_{i=1}^{n} \mathcal{L}(C^{100}(f_\theta(x_i)), y_i^{100}) \tag{11}$$

$$\mathcal{L}_{mg2} = \sum_{i=1}^{n} \mathcal{L}(C^{1,000}(f_\theta(x_i)), y_i^{1,000}) \tag{12}$$

$$\mathcal{L}_{mg3} = \sum_{i=1}^{n} \mathcal{L}(C^{10,000}(f_\theta(x_i)), y_i^{10,000}) \tag{13}$$

where $f_\theta$ is the molecular encoder, and $C$ are task-specific fully-connected neural networks for clustering label prediction. In the optimization problem in the first level of TapWeight, $C$ is optimized as well as $\theta$. However, $C$ is not relevant to the optimization problems in the other levels and is discarded after TapWeight training, so we did not explicitly include it in Equation 2.

**Mask-based Contrastive Learning**   In this pretraining objective, we use a $16 \times 16$ square area to randomly mask a molecular image $x$ to generate the masked image $\hat{x}$. We then perform constrastive learning on the image pair $(x, \hat{x})$ by minimizing the distance between representations of both images to promote consistency. Formally,

$$\mathcal{L}_{mcl} = \sum_{i=1}^{n} ||f_\theta(x_i), f_\theta(\hat{x}_i)||_2 \tag{14}$$

where $||\cdot||$ denotes the Euclidean distance between two molecular representation generated from the encoder.

**Jigsaw Puzzle Prediction**   In this pretraining objective, we introduce 100 types of different permutations with number 1 to 100, denoted as $y^{jig}$. We also assign a label of 0 for original molecular image without any We apply the permutation to molecular images $x$ to get permuted ones $\hat{x}$. The encoder $f_\theta$ is pretrained by predicting the permutation label. Formally,

$$\mathcal{L}_{jpp} = \sum_{i=1}^{n} \mathcal{L}(C(f_\theta(\hat{x}_i)), y_i^{jig}) \tag{15}$$

where $C$ is a task-specific fully-connected neural network for permutation label prediction.

## C.2  Datasets

We use the datasets from MoleculeNet benchmark for molecule property prediction Wu et al. (2017).

**Quantum Mechanics** Qm7 and Qm9 are both molecular datasets for regression task on quantum mechanics properties of molecules. Qm7 dataset collects electronic properties of molecules determined using ab-initio density functional theory (DFT). Qm9 dataset collects geometric, energetic, electronic and thermodynamic properties of DFT-modelled small molecules.

**Physical Chemistry** Esol, FreeSolv and Lipophilicity (Lipo) are all datasets for regression task on physical chemistry properties of molecules. ESOL dataset collects water solubility data for common organic small molecules. FreeSolv dataset collects experimental and calculated hydration free energy of small molecules in water. Lipo dataset collects experimental results of octanol/water distribution coefficient.

**Biophysics** Bace, HIV and MUV are all datasets for classification tasks on biophysics properties of molecules. BACE dataset collects binary label of molecular binding results for a set of inhibitors of human $\beta$-secretase 1 (BACE-1). HIV dataset collects experimentally measured abilities of a molecule to inhibit HIV replication. MUV is a subset of PubChem BioAssay by applying a refined nearest neighbor analysis, designed for validation of virtual screening techniques.

**Physiology** BBBP, Clintox, Sider, Toxcast and Tox21 are all datasets for classification tasks on physiology properties of molecules. BBBP dataset contains binary labels of blood-brain barrier penetration (permeability) ability for molecules. ClinTox dataset consists of qualitative data of drug molecules approved by the FDA and those that have failed clinical trials for toxicity reasons. Sider is a database of marketed drugs and adverse drug reactions (ADR), grouped into 27 system organ classes. ToxCast dataset contains toxicology data for a large library of compounds based on in vitro high-throughput screening, including experiments on over 600 tasks. Tox21 dataset collects qualitative toxicity measurements of molecules on 12 biological targets, including nuclear receptors and stress response pathways.

### C.3 Baseline Methods

**Attribute Masking** Attribute masking (AttrMask) based pretraining captures domain knowledge by learning the regularities of the node/edge attributes distributed over graph structure (Hu et al., 2020). Inspired by BERT (Devlin et al., 2019), it pretrains a graph neural network (GNN) by first masking node/edge attributes and then letting GNNs predict those attributes based on neighboring structure.

**Context Prediction** Context Prediction uses subgraphs to predict their surrounding graph structures (Hu et al., 2020). It pretrains a GNN so that it maps nodes appearing in similar structural contexts to nearby embeddings. Specifically, the method first encodes the context into a fixed vector using an auxiliary GNN, and then trains the GNN encoder with negative sampling.

**GraphMVP** The Graph Multi-View Pre-training (GraphMVP) framework applies self-supervised learning (SSL) by utilizing the correspondence and consistency between 2D topological structures and 3D geometric views (Liu et al., 2022). It introduces a novel contrastive learning loss, using the 2D and 3D representations of the same molecule as positive pairs.

**Imagemol** ImageMol is an unsupervised pretraining deep learning framework pretrained on 10 million unlabelled drug-like, bioactive molecules, to predict molecular targets of candidate compounds (Zeng et al., 2022). The ImageMol framework is designed to pretrain chemical representations from unlabelled molecular images on the basis of local and global structural characteristics of molecules from pixels.

### C.4 Hyperparameter Settings

We set the number of clusters for the loss terms $\mathcal{L}_{mg1}$, $\mathcal{L}_{mg2}$, and $\mathcal{L}_{mg3}$ to 100, 1,000, and 10,000, respectively. During the continued pretraining, we set the unrolling step in the MLO framework to be 1. We use the SGD optimizer with a step learning rate scheduler across all three optimization levels. All experiments are conducted on 1 NVIDIA A100 GPU.

**Classification**  We set the global learning steps to be 30,000 for MUV dataset, 20,000 for HIV dataset, 10,000 for Tox21 and Toxcast datasets, and 3,000 for all other datasets. We set the batch size in level I to be 1024, and that in level II and level III to be 64 for all datasets. We set the learning rate to be 0.02 in level I, 0.05 in level II, and that in level III to be 200 for all datasets. We set the $\gamma$ value in Equation 3 to be 0.001.

**Regression**  We set the global learning steps to be 30,000 for qm9 dataset and 10,000 for all other datasets. The batch size and $\gamma$ are the same as those in classification tasks. We set the learning rate to be 0.02 in level I, 0.001 in level II, and 1 in level III for Lipo, Esol and FreeSolv datasets. We set the learning rate to be 0.01 in level I, 0.0001 in level II, and 0.1 in level III for Qm7 and Qm9 datasets.

## D  Natural Language Understanding

### D.1  Pretraining Objectives

We use 3 types of losses for continued pretraining of an RoBERTa model Liu et al. (2019c) to enhance its performance on natural language understanding tasks.

**Mask Language Modeling**  This pretraining objective randomly mask some percentage of the input tokens, and then predict those masked tokens using embedding generated from the pretrained model (Devlin et al., 2019). In BERT and RoBERTa, 15% of the tokens are masked in the pretraining stage.

**Constrastive Learning**  This pretraining objective applies dropout noise to the encoder $f_\theta$ when taking in a sentence $x$ to get a negative sample of encoding $h' = f_\theta(x)$ (Gao et al., 2021). We use $h$ to denote those positive encodings without dropout noise. The encoder is then trained by minimizing a constrastive learning loss:

$$\mathcal{L}_{cl} = -\sum_{i=1}^{n} \log(\frac{e^{sim(h_i,h_i')}}{\sum_{j=1}^{n} e^{sim(h_i,h_j')}}) \tag{16}$$

where $sim$ is a similarity measure between two encodings.

**Sentence Order Prediction**  This pretraining objective uses two consecutive segments from the same document as positive examples. It generates negative examples using the same two consecutive segments but with their order swapped (Lan et al., 2020). The model is pretrained by predicting the label of these two types of examples.

### D.2  Datasets

We use 8 datasets from GLUE benchmark in natural language understanding tasks (Wang et al., 2019). Following standard practices, we use the original GLUE development set as the test set in our experiments, and randomly split the original training set into a training set and validation set with a ratio of 8:1.

**Single Sentence Tasks**  The Corpus of Linguistic Acceptability (CoLA) contains English acceptability judgments sourced from books and journal articles on linguistic theory. The Stanford Sentiment Treebank (SST-2) features sentences from movie reviews annotated by humans for sentiment analysis.

**Similarity and Paraphrase Tasks**  The Microsoft Research Paraphrase Corpus (MRPC) is a dataset of sentence pairs extracted from online news sources, annotated by humans for semantic equivalence. The Quora Question Pairs (QQP) dataset includes question pairs from the Quora website, where the task is to determine if the questions are semantically equivalent. The Semantic Textual Similarity Benchmark (STS-B) contains sentence pairs from news headlines, video and image captions, and natural language inference datasets, with the task of predicting a human-annotated similarity score.

**Inference Tasks**   The Multi-Genre Natural Language Inference Corpus (MNLI) is a crowdsourced dataset of sentence pairs annotated for textual entailment, where the task is to predict the relationship between a premise and a hypothesis. Question-answering Natural Language Inference (QNLI) involves question-paragraph pairs, with the task of determining whether the paragraph contains the answer to the question. The Recognizing Textual Entailment (RTE) datasets consist of sentence pairs from news and Wikipedia, where the task is to predict the entailment between two sentences.

### D.3   Baseline Methods

**RoBERTa**   The Robustly Optimized BERT Pretraining (RoBERTa) paper (Liu et al., 2019c) thoroughly evaluates the impact of key hyperparameters and training data size in BERT. RoBERTa uses the same architecture as BERT but is pretrained with an optimized strategy, leading to significant improvements in performance across various downstream tasks. The main differences between RoBERTa and BERT are: (1) training for a longer duration with larger batches and more data; (2) removing the next sentence prediction objective; (3) training on longer sequences; and (4) dynamically adjusting the masking patterns applied to the training data.

**SimCSE**   The Simple Contrastive Learning of Sentence Embeddings (SimCSE) framework includes both unsupervised and supervised approaches. In the unsupervised approach, SimCSE takes an input sentence and predicts the same sentence using a contrastive objective, where standard dropout serves as the noise. In the supervised approach, it integrates annotated pairs from natural language inference datasets into the contrastive framework, using human-labeled "entailment" pairs as positive examples and "contradiction" pairs as hard negatives.

**DeBERTa**   The Decoding-enhanced BERT with Disentangled Attention (DeBERTa) model (He et al., 2021b) introduces architectural improvements over BERT to enhance contextualized representations. De-BERTa differs from BERT primarily by incorporating (1) a disentangled attention mechanism that separately encodes content and positional information, insteading of adding both word (content) embedding and position embedding together as that in BERT; and (2) an absolute position encoding correction to improve position-dependent generalization. These enhancements enable DeBERTa to achieve superior performance on a wide range of natural language understanding tasks.

### D.4   Hyperparameter Settings

When applying TapWeight on the RoBERTa encoder, we set the unrolling step in the MLO framework to 1. We use an Adam optimizer with a step learning rate scheduler across all three optimization levels. All experiments are conducted on 1 NVIDIA A100 GPU. We set the global learning steps to 20,000 for the QQP and MNLI datasets, and 10,000 for all other datasets. The batch size for level I is set to 512, while for levels II and III, it is set to 32 across all datasets. The learning rate for levels I and II is 2e-5, and for level III, it is set to 1 for all datasets. We set the $\gamma$ value in Equation 3 to be 0.005.

## E   Selection of Proximal Regularization Loss

In this section, we discuss the selection of proximal regularization loss $\mathcal{R}$ in the second level of TapWeight, as specified in Equation 3. In principle, both mean squared error (MSE) and Kullback–Leibler divergence (KLDiv) can be used as the proximal regularization loss in Level II of TapWeight. However, we choose MSE following existing works (Rajeswaran et al., 2019; Choe et al., 2023c), where MSE is consistently used as the proximal regularization loss when converting iterative optimization in bi-level optimization problems into implicit optimization. In preliminary experiments, we observed that the MSE loss offers improved computational efficiency and numerical stability compared to the KLDiv loss, particularly in large-scale settings where both $\theta^*$ and $\omega$ comprise billions of parameters. This makes MSE a more practical choice for training at scale. To quantify the efficiency gap, we benchmarked the forward and backward computation times of MSE and KLDiv losses $\mathcal{R}(\omega, \theta^*)$. We use a RoBERTa-base backbone (approximately

| $\mathcal{R}$ | Forward | Backward |
|---|---|---|
| **KLDiv** | 0.385 | 0.484 |
| **MSE** | **0.101** | **0.108** |

Table 10: Average wall-clock times (in seconds) of forward and backward operation for MSE and KLDiv loss.

125M parameters) for $\omega$ and $\theta^*$, which is implemented in PyTorch on the same hardware (NVIDIA A100). Average wall-clock times (in seconds) are reported in Table 10.

As shown in Table 10, the MSE is approximately 4 times faster than KLDiv on both forward and backward passes, highlighting its computational efficiency. That said, it is possible that KLDiv has advantages in certain scenarios, which we aim to explore in future work.

