# OpenReview forum: "TapWeight: Reweighting Pretraining Objectives for Task-Adaptive Pretraining"
_TMLR — Accepted by TMLR_

### Review · Reviewer_KqG3 · 2025-04-12

**Summary Of Contributions:**

The paper focuses on the problem of determining optimal weights across multiple continued pre-training losses, where model performance is measured after further fine-tuning on downstream tasks. The a multi-stage approach is proposed, where the weights are initially fixed during continued pre-training, downstream fine-tuning, and evaluation on the validation set. The weights are then adjusted based on evaluation performance by differentiating through the multi-stage process. The proposed method is evaluated on two major domains: molecular property prediction and natural language processing. Results on both domains show the proposed method, TapWeight, improves over existing baseline methods for continued pre-training. The authors also conducted ablations of several aspects of TapWeight to determine their importance in improving performance.

**Audience:**

Yes

**Broader Impact Concerns:**

None.

**Claims And Evidence:**

Yes

**Requested Changes:**

1. [Critical] Section 3.1 "In our framework, TapWeight, we aim to automatically search for the optimal ... so that the pretrained model will achieve highest performance on Dval after finetuned on a downstream dataset Df t." This is the wrong objective. Surely you mean achieve highest **test** performance? Please confirm that the results reported in Section 4 are conducted on the test splits?
2. [Critical] Add an Algorithm listing of TapWeight, clearly specifying when and how many times each stage is done, the number of $\lambda$ iterations, when evaluations on downstream validation set is done, when gradients are calculated and with which algorithm, etc. Define what a "global training step" means. It should be clear enough that a reader can quickly grasp the computational complexity of TapWeight.
3. [Strengthen] The paper write up starts to describe TapWeight in "stages" but switches to "levels" half way through the paper. Please be consistent throughout and be careful with your choice of terminology. "Stages" implies that many gradient updates are done during continued pre-train, and fine-tuning, then repeat, whereas "levels" implies all of the above are done per gradient update.
4. [Critical] Specify computational complexity of the algorithm used in 3.3.
5. [Strengthen] Add computational complexity comparison with baselines.
6. [Strengthen] Discuss how TapWeight computational complexity scales with model size.
7. [Strengthen] B.1 equations (1-3), is the model under training $f_{\theta}$ or does it include the C's as well?

**Strengths And Weaknesses:**

Strengths:

1. The proposed method is novel and achieves best performance among baselines included in the paper.
2. Generality of the method across two quite different application domains (molecular property and NLP).
3. Ablations show the proposed components of TagWeight are important in achieving good performance.
4. Quality of paper presentation is high.

Weaknesses:

1. Proposed method involves differentiating through a multi-stage process. This can become very expensive computationally or becomes prohibitive for larger models. Section 4.5 attempts to compare (wall-?) time of different versions of TapWeight, but not with other baselines. Also 4.5 does not address scaling of algorithmic complexity with model size.
2. Description of the algorithm is not completely clear. How many iterations of \lambda are done? Is it every learning batch? Does this mean the method conducts an evaluation on the downstream task's validation task every gradient step?
3. Minor clarifications in writing. See "Requested Changes".

---

> ### Author Response · Authors · 2025-04-16
> **Author Response 1**
>
> We appreciate the reviewers’ recognition of our work and the valuable suggestions. Below is our detailed, point-by-point response:
>
> ### 1. Wall Time Cost for Additional Baselines
>
> We apologize for any confusion. To clarify, the methods listed in Table 6—excluding TapWeight—are baseline methods rather than variants of TapWeight. For example, the **FT** baseline in Table 5 corresponds to the **Finetuning** baseline presented in Tables 3 and 4, while the **Imagemol** baseline appears in Tables 1 and 2. We originally selected these two baselines for comparison because they apply across both modalities, whereas other baselines such as AttrMask and PCP are modality-specific and cannot be applied on other modalities. In this revision, we now include the wall time consumption for two additional baselines, **TAPT** and **PCP**, as summarized below:
>
>
> |       **Method**        | **FT** | **TAPT** | **PCP** | **TapWeight** |
> |---------------|--------|----------|--------|---------------|
> | **QQP** | 1.00   | 1.57     | 2.26   | 3.76          |
>
>
> ### 2. Complexity of TapWeight Optimization Algorithm
>
> Thank you for your suggestions regarding the complexity analysis. In summary, our optimization algorithm scales linearly with the number of model parameters, owing to the approximation of the best-response Jacobian matrix-vector multiplication, inspired by [1] and [2]. We outline our analysis below:
>
> Assume the number of optimizable parameters in the model is $n$. Directly computing the full derivative in Equation (6) would incur a complexity of $O(n^3)$, because the best-response Jacobian matrix requires computing an inverse Hessian-vector product (iHVP), and directly inverting a Hessian itself is $O(n^3)$. To address this, we leverage the Neumann series approximation to reduce the complexity to $O(n^2)$, using the following formula inspired by paper [1]:
>
> $
> v \cdot \bigl (\nabla^2_w \mathcal L(w)\bigr)^{-1}
> \approx v \cdot
> \sum_{j=0}^M
> \bigl(I - \nabla^2_w \mathcal L(w)\bigr)^j
> $
>
> where $M$ is a hyperparameter in practice. However, if we directly compute the Hessian above, the resulting quadratic complexity is still computationally intensive for models with billions of parameters. Inspired by paper [2], we further reduce the complexity by leveraging the Hessian-vector product (HVP) and finite differences. HVP has a complexity of $O(n)$ as implemented in modern automatic differentiation (AD) libraries (e.g., `torch.autograd.functional.hvp`), which is more efficient than directly computing the Hessian. A detailed algorithm for computing the best-response Jacobian-vector product can be found in Appendix B ("Practical Implementation of Hypergradient") of [2], where only a limited number of first-order gradient computations are required. Since gradient computation is $O(n)$ in modern AD libraries, the total complexity of multiplying the first two terms in Equation (6) is also $O(n)$. The resulting vector is then multiplied by a subsequent term, which can also be resolved similarly in $O(n)$ time. Overall, computing the right-hand side (RHS) of Equation (6) has a complexity of $O(n)$.
>
> Other components of our optimization algorithm—including loss computation, gradient backpropagation, and parameter updates—also operate in $O(n)$ time. Therefore, the total complexity of the TapWeight optimization algorithm grows linearly with the model size $n$.
>
> [1] Jonathan Lorraine, Paul Vicol, David Duvenaud. *Optimizing Millions of Hyperparameters by Implicit Differentiation*. AISTATS 2020.
> [2] Miao Zhang, Steven Su, Shirui Pan, Xiaojun Chang, Ehsan Abbasnejad, Reza Haffari. *iDARTS: Differentiable Architecture Search with Stochastic Implicit Gradients*. ICML 2021.

---

> ### Author Response · Authors · 2025-04-17
> **Author Response 2**
>
> ### 3. Detailed Description of the TapWeight Optimization Algorithm
>
> We appreciate the reviewer’s feedback regarding this point.  Based on the reviewers’ suggestions, we first present the detailed optimization algorithm for TapWeight.
>
> ---
>
> >**Algorithm:** `TapWeight-optimization`
> >**Input:** Unsupervised pretraining dataset $\mathcal D_{pt}$,  Training dataset $\mathcal D_{tr}$ and validation dataset $\mathcal D_{val}$ for the target downstream task.
> >**For** $i = 1$ to $M$ **do**:
> >1. Sample mini-batch $X_{pt}, X_{tr}, X_{val}$ from $\mathcal D_{pt}, \mathcal D_{tr}, \mathcal D_{val}$
> >2. Compute gradient:  $\frac{\partial \mathcal L_1}{\partial \theta}$, where $\mathcal L_1 = \mathcal L_{pt}(\theta, \lambda, X_{pt})$ (Equation 2)
>   >Update:  $\theta \leftarrow \theta - \alpha_1 \cdot \frac{\partial \mathcal L_1}{\partial \theta}$
> >3. Compute hyper-gradient: $\frac{d \mathcal L_2}{d \omega}$,  where $\mathcal L_2=\mathcal L_{tr}(\omega,X_{tr})+\gamma\mathcal R(\omega,\theta^*(\lambda))$  (Equation 3)
>   >Update: $\omega \leftarrow \omega - \alpha_2 \cdot \frac{d \mathcal L_2}{d \omega}$
> >4. Compute hyper-gradient:  $\frac{d \mathcal L_3}{d \lambda}$,  where $\mathcal L_3=\mathcal L_{val}(\omega^*(\lambda), X_{val})$ (Equation 4)
>   >Update:  $\lambda \leftarrow \lambda - \alpha_3 \cdot \frac{d \mathcal L_3}{d \lambda}$
> >
> >**Return:** Optimal weights $\theta^*$, $\omega^*$, $\lambda^*$
>
> ---
>
> One global training step denotes a single iteration in the algorithm above, and there are $M$ global steps in total. We use the term "global" to distinguish it from the three separate parameter updates that occur within a single global step. As shown in the algorithm `TapWeight-optimization`, each global training step consists of a single gradient descent update for each of the optimization problems (OPs) at all three levels. Therefore, in response to the questions from the reviewer, TapWeight evaluates the downstream task’s validation performance in every global training step, and the parameter $\lambda$ is updated accordingly.  For gradient computations in **Level I**, we use standard automatic differentiation libraries. For **Levels II and III**, we compute hypergradients based on the best-response Jacobian-vector product approximation method, as detailed in Section 2 of this response.
>
>
> ### 4. Clarification of Evaluation Splits
>
> We apologize for the earlier confusion. We confirm that all results reported in Section 4 are based on the test splits. In Section 3.1, we previously referred to $\mathcal{D}_{val}$ in the context of TapWeight’s training process, where it was used to optimize the trade-off weights $\lambda$. To avoid confusion, we have revised the phrasing to:
>
> ---
>
> >"...so that the pretrained model achieves the highest performance on the test split $\mathcal D_{ts}$ after being fine-tuned on the downstream dataset $\mathcal D_{tr}$."
>
> ---
>
> as suggested by the reviewer.
>
> ### 5. Additional Clarifications
>
> **Terminology ('Stages' and 'Levels') in TapWeight:**
> We apologize for the confusion and have now standardized the use of 'level' throughout the revised manuscript. In addition, we clarify that TapWeight performs updates iteratively across each level, with one gradient descent step per level, which aligns with the reviewer's interpretation of the term.
>
> **Complexity Comparison with Baselines:**
> All baseline methods analyzed in our study exhibit $O(n)$ complexity, which scales linearly with model size $n$. As shown in Section 2, TapWeight also maintains $O(n)$ complexity, though it incurs higher constant factors due to hyper-gradient computations. Nonetheless, our experiments demonstrate that these costs remain acceptable in most practical scenarios.
>
> **Parameters Optimized in Appendix B.1:**
> To clarify, both $C$ and $f_{\theta}$ are optimized in Level I. However, the parameters of $C$ are not involved in the optimization objectives of Levels II and III.
>
> We sincerely thank the reviewers again for their insightful and constructive feedback. Following TMLR submission guidelines, we will upload a revised manuscript incorporating our response to all comments once we have received feedback from all reviewers.

---

### Review · Reviewer_cC8s · 2025-04-18

**Summary Of Contributions:**

Given a target downstream task, TapWeight attempts to find the optimal weight allocation across different pretraining objectives. In the first stage of a tri-level optimization problem, the weighted pre-training loss is minimized. In the second stage, using another set of model parameters, performance on the downstream task (training set) is optimized, with a regularization term that encourages these other parameters to remain close to those obtained by pretraining. In the third stage, to minimize the downstream task validation loss, the pre-training objective weights are updated using the chain rule and some Jacobian approximations. The approach is evaluated on molecular property and natural language tasks, using respectively 5 and 3 pre-training objectives.

**Audience:**

Yes

**Claims And Evidence:**

Yes

**Requested Changes:**

**Critical**
- Clarify the optimization procedure (maybe with pseudo-code?).

**Would strengthen**
1.  Revise the paper to ensure it can be more easily understood by readers from different backgrounds.
2. Did you evaluate downstream performance using a single pretraining objective at a time ($\lambda_i=1, \lambda_{j\neq i}=0$)?
    - Did you try using a few random $\lambda$s?
3. What are the results if you learn the final weights, then run continued pretraining and finetune the model (starting from the initial pretrained parameters)?
4. Do you impose a constraint on the sum or norm of $\lambda$? If not, would multiplying $\lambda$ by a constant lead to the same results?
5. Could you use the KL divergence to measure how close $\theta^*$ and $\omega$ are? If yes, then why did you decide to use the parameter mean squared error? Otherwise, why not?

**Strengths And Weaknesses:**

**Strengths**

- The paper proposes an efficient optimization approach to update the pre-training objective weights $\lambda$, even if the exact gradients cannot easily be directly computed.
- The method is evaluated on a variety of tasks from 2 highly different domains (molecular properties and NLP), with generally positive results compared to other baselines and pre-training with equal weights.

**Weaknesses**

- The description of the optimization procedure was unclear to me. In particular, based on section 3, I initially thought that pre-training was run to convergence to find $\theta^*(\lambda)$, then task-specific training was completed to find $\omega^*(\theta^*(\lambda))$, and then the pretraining weights were updated based on the full validation loss, with this approach repeated for a few iterations. However, based on figure 2, I now believe that batches from the 3 stages are interleaved, but I am not sure if that is correct.
    - "three end-to-end stages" was confusing.
- Some of the baseline models are arguably outdated. It would be nice to show if the proposed approach could also be useful for more recent open-weight large language models.
- "Scaffold splitting" is introduced without a definition or citation. Some readers, especially those from a mainly NLP background, may not understand what it refers too. In general, acronyms should ideally be spelled out at least once.

---

> ### Author Response · Authors · 2025-04-22
> **Author Response 1**
>
> We appreciate the reviewers’ recognition of our work and the valuable suggestions. Below is our detailed, point-by-point response:
>
> ### 1. Detailed Description of the TapWeight Optimization Procedure
>
> We apologize for the confusion regarding the description of the optimization procedure. In response, we first present the detailed optimization algorithm for TapWeight.
>
> ---
>
> >**Algorithm:** `TapWeight-optimization`
> >**Input:** Unsupervised pretraining dataset $\mathcal D_{pt}$,  Training dataset $\mathcal D_{tr}$ and validation dataset $\mathcal D_{val}$ for the target downstream task.
> >**For** $i = 1$ to $M$ **do**:
> >1. Sample mini-batch $X_{pt}, X_{tr}, X_{val}$ from $\mathcal D_{pt}, \mathcal D_{tr}, \mathcal D_{val}$
> >2. Compute gradient:  $\frac{\partial \mathcal L_1}{\partial \theta}$, where $\mathcal L_1 = \mathcal L_{pt}(\theta, \lambda, X_{pt})$ (Equation 2)
>   >Update:  $\theta \leftarrow \theta - \alpha_1 \cdot \frac{\partial \mathcal L_1}{\partial \theta}$
> >3. Compute hyper-gradient: $\frac{d \mathcal L_2}{d \omega}$,  where $\mathcal L_2=\mathcal L_{tr}(\omega,X_{tr})+\gamma\mathcal R(\omega,\theta^*(\lambda))$  (Equation 3)
>   >Update: $\omega \leftarrow \omega - \alpha_2 \cdot \frac{d \mathcal L_2}{d \omega}$
> >4. Compute hyper-gradient:  $\frac{d \mathcal L_3}{d \lambda}$,  where $\mathcal L_3=\mathcal L_{val}(\omega^*(\lambda), X_{val})$ (Equation 4)
>   >Update:  $\lambda \leftarrow \lambda - \alpha_3 \cdot \frac{d \mathcal L_3}{d \lambda}$
> >
> >**Return:** Optimal weights $\theta^*$, $\omega^*$, $\lambda^*$
>
> ---
>
> As shown in the algorithm above, we confirm that the data batches from the three stages are indeed interleaved. We have also revised the phrase “three end-to-end stages” to “three optimization steps”, following the reviewers’ suggestion. We have incorporated the algorithm description above into the revised manuscript to clearly show the exact optimization procedure of TapWeight and to avoid any confusion.
>
> ### 2. Experimental Results with Recent Pretrained LMs
>
> We thank the reviewer for the helpful suggestion. In response, we have applied TapWeight to a more recent pretrained LM, DeBERTa-v3-large, which was published at ICLR 2023 [1]. Specifically, we compared TapWeight to direct fine-tuning of DeBERTa-v3-large on both the CoLA dataset and the MRPC dataset from the GLUE benchmark, with the results shown below.
>
> | **Task**     | **CoLA** | **MRPC** |
> |--------------|---------|----------|
> | **FT**       | 73.0    | 91.5     |
> | **TapWeight**| **74.1**    | **92.1**     |
>
>
> We observe that TapWeight still clearly outperforms the baseline method on both datasets, highlighting its generalizability to more recent pretrained LMs.
>
> ### 3. Definition of Scaffold Splitting
>
> We apologize for the confusion. In response, we have added the following line to our manuscript to explain the definition of "scaffold splitting":
>
> ---
>
> >"Scaffold splitting partitions molecules based on their core structural frameworks to ensure that structurally similar compounds do not appear across training and test sets, thereby providing a more rigorous evaluation of model generalization."
>
> ---
>
> We have also thoroughly revised our manuscript to ensure that all terminologies and acronyms are explained or spelled out at least once, making it more accessible to readers from different backgrounds.
>
> ### 4. Results with Different Selection of $\lambda$
>
> Thank you for the insightful question. We have actually compared TapWeight to both baselines mentioned by the reviewer in our manuscript, and we elaborate below.
>
> **Single Pretraining Objective:** In Table 3 and Table 4, the SimCSE baseline uses a single pretraining objective (a contrastive learning objective) for continued pretraining. TapWeight consistently outperforms this baseline across all datasets, demonstrating the importance of leveraging multiple pretraining objectives rather than relying on only one.
>
> **Random Weights for Pretraining Objectives:** In Table 5, the "CP w/o Reweighting" ablation baseline corresponds to the method that assigns random weights to pretraining objectives. Results show that TapWeight outperforms this baseline on the MoleculeNet benchmark, highlighting the advantage of dynamically reweighting pretraining objectives during the continued pretraining process.
>
> We have revised our manuscript to clearly describe both baselines and highlight the advantages of TapWeight over these baselines to avoid any confusion.

---

> ### Author Response · Authors · 2025-04-22
> **Author Response 2**
>
> ### 5. Continued Pretraining with Learned Final Weights
>
> We thank the reviewer for the insightful suggestion. To address this, we conducted additional experiments on three MoleculeNet datasets (Bace, Clintox, and HIV) to evaluate the performance of continued pretraining (CP) using the tradeoff weights identified by TapWeight (CP-final-weights), with results shown below.
>
> | **Task**     | **BACE** | **Clintox** | **HIV** |
> |--------------|---------|----------|----------|
> | **CP-final-weights**       | 82.3    | 81.1 |**78.5** |
> | **TapWeight**| **83.1** | **81.3**  |78.4  |
>
> On average, this baseline achieved a score of 80.6, which is comparable to TapWeight’s score of 80.9, demonstrating the effectiveness of the tradeoff weights found by TapWeight. However, this approach incurs additional computational cost due to an extra round of CP, without yielding significant performance gains. Therefore, it is more practical to directly use the model weights obtained from TapWeight without conducting an additional CP stage.
>
>
>
> ### 6. Constraints on the Sum of $\lambda$
>
> We apologize for the confusion. In our implementation (as shown in our code in the Supplementary Material), we do apply a constraint on these weights such that the sum of all weights is 1 ($\sum_i \lambda_i = 1$). This is achieved by reparameterizing $\lambda_i$ as $\frac{e^{\beta_i}}{\sum_j e^{\beta_j}}$ using a softmax function, and optimizing $\beta$ instead of directly optimizing $\lambda$. This inherently ensures that the sum of all weights $\lambda$ equals 1. We have revised our manuscript accordingly to describe the constraint on $\lambda$ and the reparameterization method used to enforce it.
>
>
>
> ### 7. MSE vs. KLDiv as Proximal Regularization Loss $\mathcal R$
>
> In principle, both mean squared error (MSE) and Kullback–Leibler divergence (KLDiv) can be used as the proximal regularization loss in Level II of TapWeight. However, we choose MSE following existing works [2,3], where MSE is consistently used as the proximal regularization loss when converting iterative optimization in bi-level optimization problems into implicit optimization. In preliminary experiments, we observed that the MSE loss offers improved computational efficiency and numerical stability compared to the KLDiv loss, particularly in large-scale settings where both $\theta^*$ and $\omega$ comprise billions of parameters. This makes MSE a more practical choice for training at scale. To quantify the efficiency gap, we benchmarked the forward and backward computation times of MSE and KLDiv losses using a RoBERTa-base model (approximately 125M parameters) implemented in PyTorch on the same hardware. Average wall-clock times (in seconds) are reported in the table below:
>
> | **Loss**     | **Forward** | **Backward** |
> |--------------|---------|----------|
> | **KLDiv**       | 0.385    | 0.484     |
> | **MSE**| **0.101**    | **0.108**     |
>
> MSE is approximately 4 times faster than KLDiv on both forward and backward passes, highlighting its computational efficiency. That said, it is possible that KLDiv has advantages in certain scenarios, which we aim to explore in future work.
>
> [1] Pengcheng He, Jianfeng Gao, Weizhu Chen, *DeBERTaV3: Improving DeBERTa using ELECTRA-Style Pre-Training with Gradient-Disentangled Embedding Sharing*, ICLR 2023.
> [2] Aravind Rajeswaran, Chelsea Finn, Sham Kakade, Sergey Levine, *Meta-Learning with Implicit Gradients*, NeurIPS 2019.
> [3] Sang Keun Choe, Willie Neiswanger, Pengtao Xie, Eric Xing, *Betty: An Automatic Differentiation Library for Multilevel Optimization*, ICLR 2023.
>
>
> We sincerely thank the reviewers again for their insightful and constructive feedback. Following TMLR submission guidelines, we will upload a revised manuscript incorporating our response to all comments once we have received feedback from all reviewers.

---

### Review · Reviewer_XrvN · 2025-04-23

**Summary Of Contributions:**

The article addresses the TAP (Task-Adaptive Pretraining) problem. The core challenges of TAP, like most multi-task problems, is to balance different pretraining objectives, in order to maximise the model's performance on the target downstream tasks.

TAP optimises the objectives' weights end-to-end; more specifically The goal is to find the optimal pretraining loss weights $\lambda$ that minimises the validation loss, for model weights obtained via two phases of training :
  * **1)** pretraining with $\lambda$-weighted objectives (in that phase, the target weights are fixed throughout pretraining)
  * **2)** finetuning model weights on the downstream tasks's training set

The key issues of the proposed approach is to keep the whole optimisation process computationally feasible, avoiding costly gradient computations. This is addressed in two ways:
  * The weights between the two pretraining phases are not the same, thus cutting the gradient flow
  * Implicit Function Theorem (IFT) to approximate Jacobians

The method is evaluated on two tasks: NLP (GLUE) and molecule  representation

**Audience:**

Yes

**Claims And Evidence:**

Yes

**Requested Changes:**

* **Critical for acceptance:** Better reference and placement compared to the multi-task related literature

* **Critical for acceptance** Table 6 includes latency results for Qm9 and MUV finetuning, however there is no finetuning results in their respective tables (1 and 2). It seems to method  finetuning is always an important baseline to report. In particular, on the GLUE benchmark we observe that finetuning is only **0.6%** worse than the proposed method but requires roughly 70% of the training time (Table 6, on QQP)

* **Strengthen the work**: Clarify what category each baseline belongs to: i.e., do they all use uniform weights ? do they optimise the weights via grid search, or something else ? In particular, on the GLUE benchmark in Table 4, I find it peculiar that all baselines underperform finetuning, although the introduction highlights the issue that "direct finetuning of the pretrained model often fails to deliver optimal results". This would suggest that either the baselines are not adequate, or that direct finetuning actually performs better than expected.

**Strengths And Weaknesses:**

### Strengths

  * The method is sound and the explanations are clear
  * The method is evaluated on two different datasets/tasks
  * The computational cost of the proposed method is discussed

### Weaknesses

  * Related work could be better discussed. In particular, it is not clear how the baselines differ from the proposed method in particular how they weigh the pretraining objectives; In addition, the task at hand seems highly related to multi-task learning which is only briefly addressed in the related work section (e.g. MetaWeighting). It  would be interesting to have a more in-depth discussion on the added value of the proposed method compared to those baselines

  * Direct Finetuning seems to be a competitive baseline. While the introduction presented direct finetuning as a "common approach which is often suboptimal", the results for finetuning are only reported in one table (GLUE) where it seems to perform well (better than all baselines but 1). This combined with the fact that finetuning seems to be more efficient than the proposed approach in terms of training time (Table 6) raises the question of how practical the approach is compared to simple finetuning.

---

> ### Author Response · Authors · 2025-04-29
> **Author Response 1**
>
> We appreciate the reviewers’ recognition of our work and the valuable suggestions. We have revised our manuscript accordingly and provide our point-by-point response below.
>
> ### 1. Related Works on Multi-task Learning
>
> We thank the reviewers for the suggestion. In response, we have added an additional related works section (Section 2.3) on multi-task learning (MTL) and further discussed the added value of TapWeight compared to existing methods.
>
> >Multi-task learning (MTL) enables models to learn multiple tasks simultaneously, promoting knowledge sharing and transfer while mitigating task conflict. Existing MTL methods can be broadly categorized into two groups. The first group focuses primarily on architecture design for parameter sharing. For example, [1] introduces two typical approaches for parameter sharing: hard sharing and soft sharing. [2] proposes using trainable linear mappings to dynamically select different combinations of activation maps for different tasks. [3] introduces a trainable router network to iteratively select functional blocks for different tasks. [4] employs soft attention modules to extract task-specific features from shared representations. Similarly, [5] also utilizes a trainable router network, as in [3], but with soft modularization that assigns probability weights to each connection between blocks.
> >
> >The second group of MTL methods is based on optimization and gradient operations. For instance, [6] formulates MTL as a multi-objective optimization problem, defining the overall objective as finding a Pareto-optimal solution. [7] proposes a gradient normalization algorithm that dynamically adjusts the gradient magnitudes of each task. [8] introduces gradient surgery, which projects a task’s gradient onto the normal plane of another task’s gradient if task conflicts exist. [9] applies meta-learning to search for optimal task weights. [10] proposes the first Bayesian formulation for gradient aggregation in MTL and develops a new optimization algorithm based on posterior estimation.
> >
> >Our method, TapWeight, shares more similarity with the second category as it also operates on gradients by reweighting each pretraining objective. However, TapWeight differs critically from all the MTL methods mentioned above, which generally assume that training and testing tasks are identical and do not consider a pretraining–finetuning scheme. In contrast, TapWeight specifically addresses the continued pretraining (CP) problem, where the MTL tasks during CP differ from those during finetuning. In this setting, optimizing solely for performance on pretraining tasks does not guarantee improved finetuning outcomes, making direct application of existing MTL methods inappropriate. To address this challenge, TapWeight introduces a novel multi-level optimization framework that searches for pretraining task weights to maximize finetuning performance, effectively overcoming the limitations of previous MTL approaches in the CP context.
>
> ### 2. Clarification of the Direct Finetuning Baseline
>
> We apologize for any confusion. Results for direct finetuning are reported in all main tables, including Tables 1, 2, 3, and 4. In Tables 1 and 2, the "Imagemol" baseline corresponds exactly to the direct finetuning baseline requested by the reviewer, which involves finetuning a pretrained Imagemol model. TapWeight results on these tables also apply the TapWeight algorithm on a pretrained Imagemol model. To reduce confusion, we have renamed the direct finetuning baseline method in these tables as "Finetuning (Imagemol)". As shown, TapWeight consistently outperforms the direct finetuning baseline. For example, TapWeight achieves an average AUROC of 75.3 compared to 73.2 for direct finetuning on 8 classification tasks in MoleculeNet benchmark. This performance gain accounts for the increased computational cost of TapWeight relative to direct finetuning.

---

> ### Author Response · Authors · 2025-04-29
> **Author Response 2**
>
> ### 3. Clarification of Baseline Settings
>
> We thank the reviewer for the suggestion. The baseline categories are listed as follows:
>
> **Finetuning without CP:** AttrMask, ContextPred, GraphMVP, Imagemol, Finetuning.
>
> **CP with a single pretraining objective:** SimCSE, TAPT, PCP.
>
> **CP with predefined weights:** CP w/o Reweighting (as shown in Table 6).
>
> **CP with searched weights:** TapWeight w/o MLO (as shown in Table 6).
>
> Existing CP baselines primarily focus on a single pretraining task and do not perform MTL continued pretraining. The MTL continued pretraining baseline with predefined weights, treated as a hyperparameter, is presented in Table 6 as an ablation study to validate the effectiveness of TapWeight's automatic tradeoff weight searching strategy. TapWeight outperforms this baseline (CP w/o Reweighting) across all datasets, demonstrating the advantage of dynamically reweighting pretraining objectives in the continued pretraining process.
>
> We did not use grid search as a baseline because the computational cost would be prohibitive. For instance, even considering only three configurations for each task weight leads to $3^5=243$ combinations when applied to Imagemol with 5 pretraining objectives, requiring an impractically large number of CP and finetuning runs. This underscores the computational efficiency of TapWeight's gradient-based search strategy.
>
> ### 4. Analysis of Continued Pretraining Baselines
> We thank the reviewer for the insightful comments. Previous CP methods, such as SimCSE, often improve performance only when the pretraining tasks or datasets are highly similar to the downstream ones. When there is significant discrepancy between CP tasks and downstream tasks, CP may negatively impact performance compared to direct finetuning. This observation further highlights the necessity of TapWeight’s dynamic reweighting mechanism, which can downplay the influence of CP tasks that are not beneficial for downstream datasets. TapWeight thus effectively addresses the limitations of prior CP approaches.
>
> [1] Sebastian Ruder, An Overview of Multi-Task Learning in Deep Neural Networks, arXiv 2017.
> [2] Ishan Misra, Abhinav Shrivastava, Abhinav Gupta, Martial Hebert, Cross-stitch Networks for Multi-task Learning, CVPR 2016.
> [3] Clemens Rosenbaum, Tim Klinger, Matthew Riemer, Routing Networks: Adaptive Selection of Non-linear Functions for Multi-Task Learning, ICLR 2018.
> [4] Shikun Liu, Edward Johns, Andrew J. Davison, End-to-End Multi-Task Learning with Attention, CVPR 2019.
> [5] Ruihan Yang, Huazhe Xu, Yi Wu, Xiaolong Wang, Multi-Task Reinforcement Learning with Soft Modularization, NeurIPS 2020.
> [6] Ozan Sener, Vladlen Koltun, Multi-Task Learning as Multi-Objective Optimization, NeurIPS 2018.
> [7] Zhao Chen, Vijay Badrinarayanan, Chen-Yu Lee, Andrew Rabinovich, GradNorm: Gradient Normalization for Adaptive Loss Balancing in Deep Multitask Networks, ICML 2018.
> [8] Tianhe Yu, Saurabh Kumar, Abhishek Gupta, Sergey Levine, Karol Hausman, Chelsea Finn, Gradient Surgery for Multi-Task Learning, NeurIPS 2020.
> [9] Yuren Mao, Zekai Wang, Weiwei Liu, Xuemin Lin, Pengtao Xie, MetaWeighting: Learning to Weight Tasks in Multi-Task Learning, Findings of ACL 2022.
> [10] Idan Achituve, Idit Diamant, Arnon Netzer, Gal Chechik, Ethan Fetaya, Bayesian Uncertainty for Gradient Aggregation in Multi-Task Learning, ICML 2024.

---

### Decision · Action_Editor_qsFZ · 2025-05-31

**Recommendation:** Accept as is

**Comment:**

This paper proposes a strategy to set task weights in pretrain/fine-tune setups by backpropagating through the pre-training, fine-tuning, and evaluation steps using the implicit function theorem. After the discussion period, all recommend or lean toward acceptance. The AE concurs with the reviewers that the paper meets TMLR's evaluation criteria.

**Audience:**

This paper focuses on task-adaptive pretraining, which is an important real-world approach to training models to accomplish tasks for which limited data is available. Although IFT-based approaches to task-adaptive pretraining have been explored in previous work, the exact recipe used here is novel, and the submission focuses on applications that the previous work did not investigate.

**Claims And Evidence:**

This paper proposes a strategy to set task weights in pretrain/fine-tune setups by backpropagating through the pre-training, fine-tuning, and evaluation steps using the implicit function theorem. It demonstrates the superiority of this appraoch over baseline methods on molecular property prediction and NLP tasks. Reviewers originally raised some concerns regarding the clarity of the method description and the strength of the baselines, but the authors addressed these concerns reviewers' satisfaction during the discussion period.